# Flash Characteristics and Precipitation Metrics of Western U.S. Lightning-Initiated Wildfires from 2017

**Brittany R. MacNamara [1,***, Christopher J. Schultz [2] and Henry E. Fuelberg [1]**

[1] Department of Earth, Ocean, and Atmospheric Science, Florida State University, Tallahassee, FL 32306, USA; hfuelberg@fsu.edu
[2] NASA George C. Marshall Space Flight Center, Huntsville, AL 35812, USA; christopher.j.schultz@nasa.gov
* Correspondence: brittany.macnamara@noaa.gov; Tel.: 1-843-747-5860

**Abstract:** This study examines 95 lightning-initiated wildfires and 1170 lightning flashes in the western United States between May and October 2017 to characterize lightning and precipitation rates and totals near the time of ignition. Eighty-nine percent of the wildfires examined were initiated by negative cloud-to-ground (CG) lightning flashes, and 66% of those fire starts were due to single stroke flashes. Average flash density at the fire locations was 1.1 fl km$^{-2}$. The fire start locations were a median distance of 5.3 km away from the maximum flash and stroke densities in the 400 km$^2$ area surrounding the fire start location. Fire start locations were observed to have a smaller 2-min precipitation rate and 24-h total rainfall than non-fire start locations. The median 2-min rainfall rate for fire-starting (FS) flash locations was 1.7 mm h$^{-1}$, while the median for non-fire-starting (NFS) flash locations was 4.7 mm h$^{-1}$. The median total 24-h precipitation value for FS flash locations was 2.9 mm, while NFS flash locations exhibited a median of 8.6 mm. Wilcoxon–Mann–Whitney rank sum testing revealed statistically different Z-Scores/*p*-values for the FS and NFS flash populations. These values were $-5.578/1.21 \times 10^{-8}$ and $-7.176/3.58 \times 10^{-13}$ for the 2-min precipitation rate and 24-h total rainfall, respectively. Additionally, 24-h and 2-min precipitation rates were statistically significantly greater for holdover versus non-holdover fire events. The median distances between the fire start location and greatest 2-min precipitation rate and greatest 24-h precipitation total were 7.4 and 10.1 km, respectively.

**Keywords:** lightning; wildfires; flash density; precipitation; holdover fires

## 1. Introduction

Lightning-initiated wildfires (LIWs) are responsible for 56% of the total acreage burned by wildfires within the continental U.S. from 1992 to 2012, resulting in an average of 2.3 million acres consumed per year [1,2]. A general characteristic of lightning flashes that ignite wildfires is a long continuing current (LCC) [3,4]. Continuing current (CC) is the length of time (longer than ~40 ms) during which charge flows through the lightning channel to the surface. The longer the CC, the more the strike location is heated, producing a greater chance of fuel ignition [5–9]. Positive cloud-to-ground (+CG) flashes have been shown to contain longer CC than negative cloud-to-ground (-CG) flashes. Thus, +CG flashes are thought to be responsible for most natural wildfires [3,6,10,11]. However, recent research indicates that 90% of lightning-initiated wildfires between 2012 and 2015 were ignited by –CG flashes [12].

The rainfall that accompanies most thunderstorms plays an important role in LIW ignition. Dry lightning, which is usually associated with LIWs, is defined as cloud-to-ground (CG) flashes that have very little to no nearby rainfall [13–15]. It is important to note that the definition of a dry lightning event varies regionally according to the United States Forest Service (USFS). The eastern and southern U.S. regions have a 24-h rainfall threshold of 6.4 mm (0.25 inch) or less, and all other regions are

2.5 mm (0.10 inch) or less [14]. Thunderstorms that produce dry lightning are most common during the summer months in the western, more arid region of the U.S. These storms create ideal conditions for wildfire ignitions since they can produce large amounts of CG lightning and strong surface winds that drive fires [1,14,16].

Current USFS metrics increase LIW probability with increasing ground flash density (e.g., number of lightning flashes per $km^2$); if the ignition efficiency (e.g., number of flashes per $km^2$ necessary to ignite a fire) is high (extreme), nine (five) flashes within a 1 $km^2$ area are expected to produce an ignition [6,17]. However, few published studies have described how flash densities can serve as a metric for wildfire prediction. The literature does suggest a relationship between lightning and cloud properties, where greater lightning flash rates are associated with a large concentration of precipitation-size ice (e.g., graupel, ice crystals), strong updrafts, and high precipitation rates [18–26]. Therefore, it has been assumed that the greatest lightning flash densities occur in the same location as the greatest rain rates.

Research has demonstrated correlations between low flash density, small flash rates, and low precipitation totals at the location of wildfire ignition [15,27]. Vant-Hall et al. [15] defined dry lightning as rainfall and rain rate values less than 0.3 mm and 0.2 mm $h^{-1}$, respectively, as a threshold for dry lightning occurrence. However, more research is necessary to demonstrate the spatial relationships between precipitation, flash density, and frequency because studies like Vant-Hall et al. [15] and Abatzouglou et al. [27] are large sample correlation analyses. Case studies and finer scale analyses are important to characterize the development of new real-time algorithms to identify LIW for impact-based decision support similar to what has been performed for other wildfire observations [28–30]. Therefore, the goals of this work are the following:

1. Characterize the flash density, polarity, 2-min rainfall rate, and 24-h precipitation total at the location of the fire-starting flash and all other flashes surrounding the fire.
2. Characterize the distance between the fire-starting lightning flash and the local maximum in precipitation, the precipitation rate at the time of the lightning flash, and the highest flash density within a 400 $km^2$ area around the fire start location.
3. Determine the frequency at which various thresholds used to define dry lightning are exceeded in the Western United States where LIW is most frequent.

## 2. Materials and Methods

### 2.1. Data

2.1.1. Fire Database and Case Selection

The wildfire database created in this study consists of LIW events from the Western U.S. that occurred between April and October 2017 and were reported in the National Wildfire Coordinating Group (NWCG)'s Incident Information System (InciWeb) [31] and the Fire Information for Resource Management System (FIRMS) [32]. InciWeb reports the cause (human vs. natural) and the approximate coordinates and ignition dates of wildfires, while FIRMS provides precise coordinates and ignition dates of wildfires derived from the Visible Infrared Imaging Radiometer Suite (VIIRS) 375-m fire product. Because FIRMS does not report the cause of the fire, InciWeb is the initial source for the Incident Type. However, FIRMS data must be used to refine InciWeb's data via precise satellite-derived fire start locations, dates, and times. The VIIRS sensor is onboard the joint NASA/NOAA Suomi National Polar-orbiting Partnership (Suomi-NPP) satellite which uses five high resolution channels to detect fires [33]. A fire is detected by the satellite sensor when the average brightness temperature of a ground pixel reaches a particular threshold [33]. FIRMS assigns a confidence level to the fire pixels (high, intermediate, or low) based on an algorithm. When the coordinates of the satellite-derived fire locations were selected for this study, only those assigned a high confidence were chosen. Low and medium confidence levels were avoided to decrease the potential for misclassifying daytime sun glint

as a wildfire [33]. The limitation of the VIIRS sensor's resolution implies potential missed micro-scale sized lightning-initiated fires that may get extinguished before they are able to expand into a sizeable fire detectable by the satellite. This caveat limits the database in this study to wildfires that are large enough to meet to required thresholds for detection.

### 2.1.2. Lightning Data

Lightning data were obtained from Vaisala's National Lightning Detection Network$^{TM}$ (NLDN), which consists of 113 stations over the contiguous U.S. [34–36]. The network records for each discharge the date, time, location, multiplicity, polarity, peak amplitude ($I_p$), and type of event (intra-cloud - IC or cloud-to-ground - CG). The NLDN has a median location accuracy of less than 0.25 km, a 90–95% CG flash detection rate [35], and stroke times that are accurate to within a few microseconds. These times and locations were used to confirm that a fire was lightning induced [35,36]. Both flash and stroke data from NLDN were used to determine the fire starter of each wildfire. An individual lightning flash can consist of either one or multiple return strokes, referred to as a single stroke flash and multi-stroke flash, respectively. In other words, a flash represents the entire discharge and a stroke represents the individual discharges within the flash. Parameters of interest were time, latitude, longitude, peak current, and multiplicity.

### 2.1.3. Precipitation Data

Archived radar data were obtained from the Multi-Radar Multi-Sensor (MRMS) data product and available through Iowa State University's public website (http://mtarchive.geol.iastate.edu) and the National Oceanic and Atmospheric Administration's National Center for Environmental Information (NOAA NCEI). The MRMS radar parameters have a grid spacing of $0.01° \times 0.01°$ (~1100 m × 1100 m), with 33 vertical levels at 2-min intervals over the continental U.S. and southern Canada [37].

The MRMS precipitation products were the 2-min surface precipitation rate (mm h$^{-1}$) and the 1-h gauge-corrected QPE (quantitative precipitation estimation; mm), i.e., the radar-derived precipitation accumulation bias corrected using local gauge amounts. The MRMS precipitation rate is derived using three different Z–R relationships depending on the type of precipitation and the geographical region [38]. The MRMS gauge-corrected QPE product is preferred over the MRMS's radar-only QPE, because it utilizes local gauge data to help mitigate errors due to limitations in the radar data.

### *2.2. Data Interrogation and Statistical Analysis*

As a first step in identifying cases, lightning-initiated wildfires via InciWeb during 2017 are selected. Then, each LIW report is paired with the NLDN data to confirm that the fire indeed was initiated by lightning. For each fire, a 20 km by 20 km domain (400 km$^2$) with 1-km grid spacing was superimposed on the reported fire location. A backward search from the reported event time was performed in this area to determine the closest lightning flash within 2 km of the start location, similar to the study by Nauslar [39]. This search of lightning data extended back as far as 10 days because 48% of LIWs have been observed to be holdover events [12] that occur when fuels ignited by lightning smolder for days or even weeks before they intensify into a detectable fire [15,22].

If no flash or stroke occurred within 10 days of the fire ignition date nor fell within 2 km of the fire ignition location, then that wildfire was discarded from the database (i.e., not further analyzed). If a single fire-starting flash or stroke was identified that met both the spatial and temporal requirements, *all* flashes and strokes within the 400 km$^2$ area of the fire and ±12 hours of the fire-starting flash were identified and placed into 1 km$^2$ bins (Figure 1a). Both flash density and stroke density (number of return strokes per km$^2$; Figure 1) were then computed separately. Flashes were separated into two categories–fire-starting (FS) flashes and non-fire-starting (NFS) flashes based on a flash's proximity to the fire-starting point. If multiple flashes were located within 2 km of the fire start location, all of them were retained as FS flashes (Figure 1b). Note it is assumed in this paper that the actual fire start location would be the same location as the flash's ground strike point rather than the satellite-derived location.

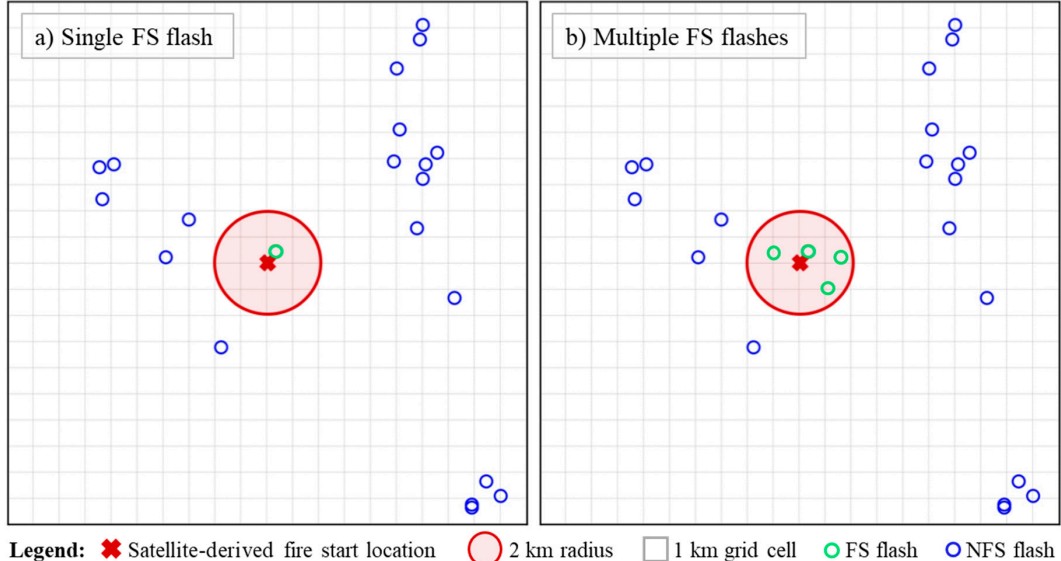

**Figure 1.** (**a**–**b**) Locations of cloud-to-ground lightning strokes within a 400 km$^2$ domain during a 2-h period. This figure illustrates a general example of a wildfire that contains (**a**) a single potential fire-starting (FS) flash and (**b**) containing multiple (four) potential FS flashes.

The flash and stroke locations were assigned to the closest 2-min precipitation rate, and 24-h QPE values (derived by totaling 1-h QPE values) to represent the total rainfall at the surface flash location. A 2.54-mm rainfall threshold during a 24-h period is currently used by the USFS to characterize dry lightning events in the Western United States [14]. It is used to indicate how many flashes meet the dry lightning criteria that do not result in a wildfire. The location of the maximum 24-h QPE value within the 400 km$^2$ grid was also identified to compute the distance between this location and the location of the FS flash in each case.

The FS and NFS flash categories were analyzed using the Wilcoxon–Mann–Whitney (WMW) rank sum test; a non-parametric test that does not require the assumption of normal distributions [40]. WMW was used to determine if the populations of fire starter flashes and non-fire starter flashes were statistically independent. The null hypothesis ($H_0$) for WMW is that the FS and NFS flash populations are similar. Because one goal of this study is to determine differences between the FS and NFS flash populations for each rainfall parameter (e.g., 2-min rainfall rate and 24-h QPE), the study seeks to reject $H_0$. Any comparison between the FS and NFS flash populations for a given parameter that exhibits $|Z - Scores| > 1.96$, indicates that $H_0$ has been rejected. That is, there are differences in the FS and NFS populations of flashes for that parameter. WMW was also used to compare 2-min rain rates of FS flashes reported within one day of a lightning occurrence to those of holdover fires. Again, $H_0$ states that the FS and NFS flash populations are similar. The goal of this exercise is to determine if sub 24-h precipitation totals can provide additional insight into the ignition time of a LIW to provide quicker identification using a future algorithm.

Because of large differences in the NFS and FS flash counts, random sub-sampling of the NFS flash data was performed to achieve near identically sized populations. The subsampling was performed by generating random arrays of 1s and 0s equal to the array size of the NFS flash database [41]. The result was an array with 100 to 150 one's digits throughout an array size of 1028 (the NFS flash sample size). This array was then multiplied by the NFS flash precipitation rate dataset to generate 100–150 randomly selected values for direct comparison with the FS flash precipitation rates. The random selection was repeated 30 times to understand the variability of the random sampling technique and provide a range of *p*-values for additional support or rejection of $H_0$. This same process was also performed on the NFS flash 24-h precipitation dataset to generate a second randomly selected array to compare with the FS flash 24-h precipitation dataset.

Another set of WMW tests were performed to determine if the maximum stroke density and maximum rain rate locations were co-located. Because flash densities typically correlate with rain rate, the test focuses on the co-location between the maximum stroke density and the maximum rain rate. Distances from the FS stroke to the maximum stroke density (distance A) were compared to the distances from the FS stroke to the maximum precipitation rate at 2-min (distance B). $H_0$, in this case, states that there is no statistical difference in the distributions between distances A and B, and the maximum stroke density locations may be co-located with the maximum precipitation rates. The alternative hypothesis ($H_1$) is that there exists a difference in the distributions of distances A and B. Thus, if this hypothesis is accepted, then the maximum stroke density and maximum precipitation rate are not co-located. Metrics were calculated to quantitatively describe Distance A, where only the fires with a stroke density maximum greater than one were included. Quantitative values of Distance B, as well as from the FS stroke to the area of maximum 24-h rainfall (Distance C), were analyzed. The area of maximum precipitation rate and 24-h rainfall was defined as the grid cell with the greatest rain rate or the grid cell containing the greatest total rainfall, respectively.

## 3. Results and Discussion

A total of 122 fires was identified between April and October 2017 using InciWeb [31]. Of these 122 events, 25 were excluded due to the inability to find either a flash or stroke that caused the fire within 2 km of the fire start location and 10 days from the report date. These exclusions may be due to the lightning occurring outside of the 2 km radius used to associate lightning to the wildfire, an incorrect classification of the cause of initiation (human instead of lightning-initiated), or they may be long duration holdover events that are beyond the 10 days used in this study as shown in the previous study by Schultz et al. [12]. In reference to NLDN data, the 95th percentile lightning distance is a 5 km median error, with the 75th percentile being ~1.6 km [42]. About 80% of LIWs in this study were within 5 km of the closest lightning point.

Note that the definition of a dry lightning event varies regionally. Because the eastern and southern U.S. have a higher 24-h rainfall threshold than the rest of the U.S., two additional fires, the West Mims and International Fires both located in Florida, were excluded. Because the West Mims fire occurred in April, this month is now excluded from the time range. Therefore, 95 LIW events occurring between May and October 2017 were analyzed in the following sections (Figure 2).

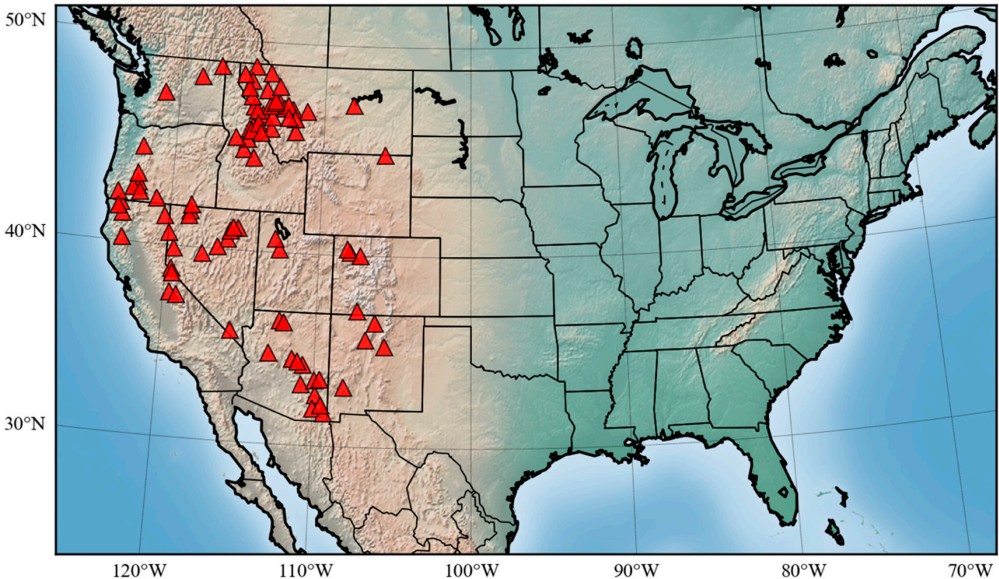

**Figure 2.** Satellite-derived locations of the 95 western U.S. lightning-initiated wildfires occurring between May and October 2017 that comprise the dataset.

### 3.1. Flash Characteristics

A total of 142 FS flashes, consisting of 294 strokes, plus an additional 1028 NFS flashes were found. The results that follow are based on flash data unless specifically stated otherwise. The polarity of the combined FS and NFS CG flashes (1170 CG flashes) is 89.7% negative. There is a clear preference for –CG fire starters. Approximately 88.7% of fire starters are –CG flashes, while only 11.3% are +CG (Table 1). Table 1 displays the median $I_p$ for both −CG fire starters and +CG fire starters of −21.1 and 30.0 kA, respectively. Similar to the FS flashes in the present study, the median $I_p$ is −17.3 kA for all −CG flashes and 23.1 for all +CG flashes (Table 1). Thus, the non-fire starters exhibit slightly weaker peak currents than their fire starter counterparts.

**Table 1.** Statistics of CG flashes during the 2017 wildfire season of this study. The characteristics include polarity, multiplicity, and peak current of FS and non-fire-starting (NFS) flashes.

| Flash Type | No. of CG Flashes | Percentage | No. of Single Stroke Flashes | Mean Multiplicity | Median $I_p$ (kA) |
|---|---|---|---|---|---|
| −CG FS | 126 | 88.7 | 59 | 3.0 | −21.1 |
| +CG FS | 16 | 11.3 | 13 | 1.2 | 30.0 |
| −CG NFS | 923 | 89.8 | 436 | 2.8 | −16.8 |
| +CG NFS | 105 | 10.2 | 94 | 1.2 | 21.5 |
| All −CG | 1049 | 89.7 | 494 | 2.8 | −17.3 |
| All +CG | 121 | 10.3 | 107 | 1.2 | 23.1 |

The frequency of stroke density at the FS flash locations is given in Figure 3a. The average stroke density is 1.7 strokes km$^{-2}$. Approximately two-thirds of the wildfires (63 of 95) ignited with a stroke density of only 1 stroke km$^{-2}$ within the standard 2-h period. Only five of the wildfires experienced ignition stroke densities ≥ 5 strokes km$^{-2}$, with four having stroke densities between 5 and 7 strokes km$^{-2}$. The extreme case in the dataset exhibited a stroke density of 13 strokes km$^{-2}$ at the location of fire ignition. This fire occurred in extreme southern Colorado where climatologically greater flash densities have been observed [43]. Finally, flash densities (Figure 3b) are smaller than stroke densities with an average value of 1.1 fl km$^{-2}$. Flash densities are expected to be less than stroke densities because multiple strokes can exist in a single flash. These results support the proposition that the fires were initiated in locations where the flash (stroke) densities are less than what the USFS current metrics utilize (i.e., ≥ 5 fl km$^{-2}$ [6,18]).

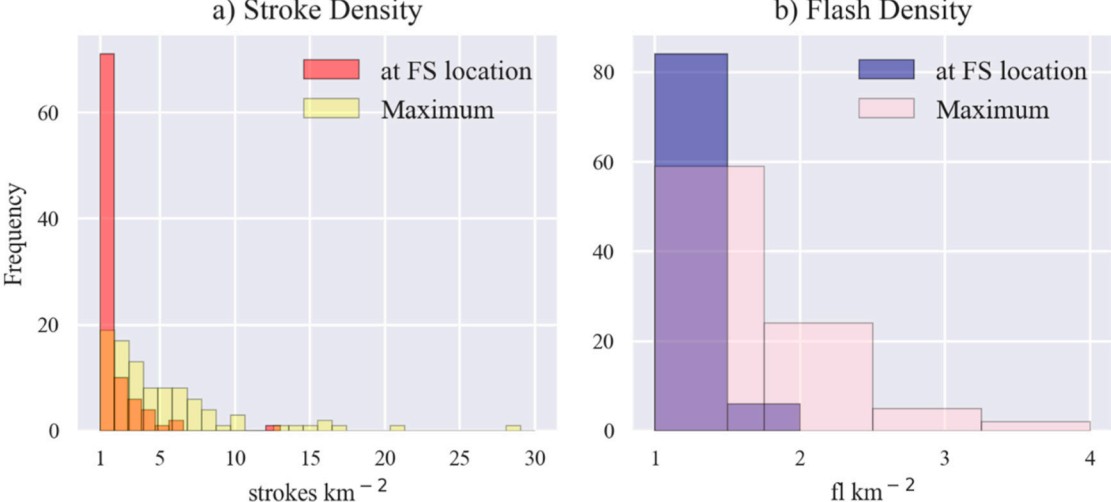

**Figure 3.** Frequency of (**a**) stroke density at the FS flash location and maximum stroke density, (**b**) mean flash density (rounded to the nearest tenth) at the FS flash location and maximum flash density.

It is also useful to examine maximum stroke density in the entire 400 km$^2$ domain to determine the maximum density near the location of each fire (Figure 3a–b). These densities were also calculated over a 2-h period. The greatest stroke density in the domain varied greatly with each fire, with a mean of 4.9 strokes km$^{-2}$ and a standard deviation of 4.8 strokes km$^{-2}$. Nonetheless, even the greatest stroke and flash densities found anywhere in the domain are typically less than the USFS thresholds of 5 and 9 fl km$^{-2}$ [6,17]. In general, smaller stroke densities exist throughout the entire domain (Figure 3b). In fact, 57 fires (60%) exhibit a maximum stroke density < 5 strokes km$^{-2}$.

The flash data reveal that 83 of the 95 wildfires had flash densities of 1 fl km$^{-2}$ within the 2-h period. Fifty-one (61.4%) of these were single stroke flashes. No wildfires exhibit a flash density ≥ 5 fl km$^{-2}$ anywhere in the domain (Figure 3b). The greatest maximum flash density is 4 fl km$^{-2}$ which occurs with only two of the fires (Figure 3b). Stroke/flash densities are examined in more detail in the case study analysis of the Parker 2 and Bridge Creek Fires.

### 3.2. Rainfall Metrics of Fire Starters and Non-Fire Starters

Precipitation at the locations of both fire starters and non-fire starters is now examined. The 2 min rainfall rate was used as the "instantaneous" metric to determine rainfall at the time of the lightning flash, while 24-h QPE was used as a longer-term measure of rainfall. The latter is useful when considering land-surface parameters and may bridge the connection between land and atmosphere.

**2-min Precipitation Rate and 24-h Rainfall Totals**

Figure 4 shows distributions of precipitation rates for both FS and NFS flashes, with the corresponding numerical data in Table A1. Approximately 96% of the fires occurred when rainfall rates were less than 20 mm h$^{-1}$. The mean rainfall rate is 4.33 mm h$^{-1}$ for FS flashes, while non-fire starters exhibit a mean rate of 12.36 mm h$^{-1}$ (Table A1). These values correspond to 0.36 mm and 1.03 mm during a standard 5-min radar scan for the FS and NFS flashes, respectively. The maximum 2-min rainfall rate for FS flashes is 43.78 mm h$^{-1}$ but is 147.40 mm h$^{-1}$ for NFS flashes. The NFS flash cases exhibit more and greater extremes than the FS cases. One should note that the mean local rain rate at the NFS flash locations is ~8 mm greater than at FS flash locations. Therefore, the statistics reveal smaller precipitation rates where FS flashes occurred.

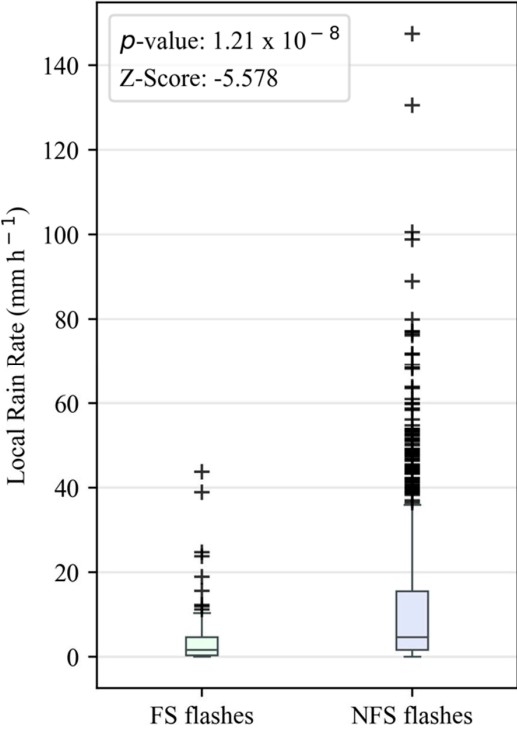

**Figure 4.** Distributions of rain rates at FS and NFS flash locations.

The distributions of FS and NFS rain rates were compared using the WMW test. Results show a Z-Score of −5.578 and a *p*-value of $1.21 \times 10^{-8}$ (Table A1). These values indicate highly statistically significant differences between the precipitation rate distributions of the FS and NFS flashes, thereby supporting the rejection of $H_0$. When the 30 random samples of the NFS flash database are compared with the FS flash database, Z-Scores range from 4.130 to 6.011 with an average of 5.203, continuing to indicate that the FS and NFS rainfall rates are from different distributions. Thus, the lightning flashes that caused the 95 LIWs during 2017 occurred in areas of smaller rainfall rates.

It is useful to compare 2-min rainfall rates for FS flashes reported within one day of lightning occurrence with those that holdover for two or more days. Separating the FS flash population into these two categories yields 65 flashes at the FS flash locations for the 0–1 day events, while the two days or more population is 77 flashes. The 25th percentile, median, and 75th percentile 2-min rainfall rates for the 0–1 day fires are 0.2 mm $h^{-1}$, 1.7 mm $h^{-1}$, and 3.3 mm $h^{-1}$, but for the 2 or more day fires are 1.4 mm $h^{-1}$, 5.1 mm $h^{-1}$, and 7.8 mm $h^{-1}$. WMW rank sum testing indicates that these two FS flash populations are statistically different, with a Z-Score of 4.045 and a *p*-value of $2.58 \times 10^{-5}$. Comparing these subcategories with the 30 random samples of the NFS flash database continues to indicate statistically significant differences between the 0–1 day FS flashes, 2–10 FS flashes, and the NFS flash populations. However, Z-Scores are much greater for the 0–1 day FS flash subcategory vs. the 2–10 day FS flash category (range of 0–1 day Z-Scores: 6.180–7.290; range of 2–10 day Z-Scores: 2.080–3.990).

Similar to the 2-min precipitation rate, the 24-h QPE data were examined to analyze total rainfall. The distributions of 24-h rainfall at the locations of FS and NFS flashes are illustrated in Figure 5, with statistical values in Table A2. Results show that the mean 24-h storm total for the NFS flash cases is almost double that of the FS flashes, 10.63 mm and 5.34 mm, respectively (Table A2). Thus, less rainfall is accumulated during the 24-h period (± 12 h from the time of fire initiation) in locations where FS flashes occurred. The range of values for FS flashes is 0.10–33.15 mm compared to 0.00–42.70 mm for the NFS flashes (Figure 5). Overall, these values differ slightly from those of Nieto et al. [44] who reported 24-h accumulated rainfall for lightning-induced fires to be less than 22.5 mm and 9 mm, depending on the region within the study. These results reflect the importance of regional differences in dry lightning criteria. Approximately 42% (60/142) of FS flashes in the present study occur during light rainfall (i.e., totals ≤ 2.54 mm during a 24-h period). Conversely, 135 (13.1%) of NFS flashes occur in areas of light rainfall.

The Z-Score for the distributions of 24-h rainfall totals for FS and NFS flash locations is −7.176, and the *p*-value is $3.58 \times 10^{-13}$ (Table A2), which rejects $H_0$. Thus, the distributions of 24-h rainfall between FS and NFS flashes are statistically different. When the 30 random samples of the NFS flash database are compared with the FS flash database, Z-Scores range from 3.656 to 6.820 with an average of 5.343, continuing to indicate that the FS and NFS flash samples are from different distributions. These findings indicate that the lightning strikes causing forest fires occur in regions where 24-h rainfall is less than in NFS flash locations.

The 24-h precipitation totals for fires reported within 1 day of lightning occurrence are now compared with those that holdover for 2 or more days. Similar to the 2-min precipitation rate, we found that statistically significant differences exist between the 0–1 day and 2 or more day 24-h precipitation. The Z-Score between the 0–1 day and 2 or more day populations is 5.290 with a *p*-value of $5.96 \times 10^{-8}$. The 25th percentile, median, and 75th percentile 24-h rainfall totals for the 0–1 day fires are 1.4 mm, 2.2 mm, and 3.8 mm, while corresponding values for fires that holdover for 2 or more days are 3.3 mm, 7.1 mm, and 10.4 mm, respectively.

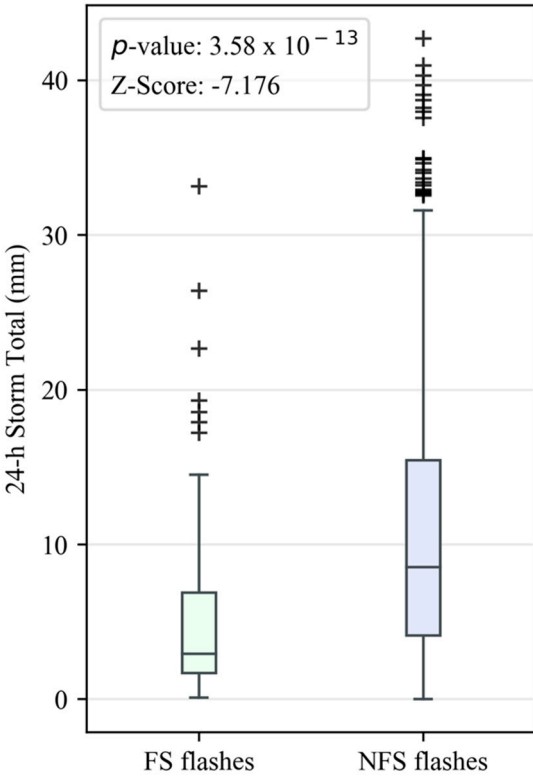

**Figure 5.** Distributions of 24-h precipitation totals at FS and NFS flash locations.

*3.3. Analysis and Quantification of Fire Initiation with Respect to Local Distances*

It is useful to determine how well the areas of maximum stroke density of FS events are co-located with their areas of maximum rain rate. This is done by using WMW to compare the distributions of distances between the FS stroke and the location of maximum stroke density (Distance A) and the distances from the FS stroke to the location of maximum rain rate (Distance B). Stroke density rather than flash density was again used because the former has more ground strike points. A statistical representation of distances from the FS strokes to the maximum stroke density (Distance A) is shown in Table 2. The average of Distance A is 5.22 km, with a median of 5.3 km and a standard deviation of 3.2 km. Of the 95 wildfires, 76 exhibit a maximum stroke density greater than 1 stroke $km^{-2}$ located within their domain. Regarding those 76 fires, 46 (60.5%) contain FS strokes located > 4 km from the maximum stroke density. Conversely, 16 fires (21.1%) have FS strokes located either at the site of maximum stroke density or < 2 km away. The Z-Score and p-value of $-3.78$ and $7.89 \times 10^{-5}$, respectively, both indicate statistically significant differences between Distances A and B; thus, rejecting $H_0$. Therefore, the areas of maximum stroke density and maximum rain rate are not co-located. Alternatively, one could speculate that because the areas of rain propagate with time, the maximum stroke density should instead be co-located with the maximum 24-h QPE. However, this speculation is not confirmed by the data. Instead, the results show that the maximum stroke density is, on average, ~2.2 km closer to the maximum rain rate than the location of greatest 24-h rainfall; 7.9 km and 10.1 km respectively. It is not unexpected that areas of maximum stroke density and maximum precipitation rate do not overlap at this scale.

**Table 2.** Distances between FS strokes and the maximum stroke density (Distance A), maximum precipitation rate (Distance B), and the maximum 24-h accumulated rainfall (Distance C).

|  | Distance A (km) | Distance B (km) | Distance C (km) |
| --- | --- | --- | --- |
| Minimum distance | 0.00 | 0.70 | 0.97 |
| 25th percentile | 2.85 | 4.66 | 4.66 |
| Median | 5.28 | 7.40 | 10.17 |
| Mean | 5.22 | 7.53 | 9.83 |
| 75th percentile | 7.66 | 10.35 | 13.00 |
| Maximum distance | 12.34 | 16.88 | 16.84 |
| Standard deviation ($\sigma$) | 3.23 | 3.64 | 4.12 |

The WMW test is used to determine if there is a difference in the distributions of stroke density at the location of fire initiation (i.e., FS stroke location) and the stroke density at the location of maximum stroke density. A Z-Score of $-7.02$ (*p*-value of $1.25 \times 10^{-13}$) reveals that the distribution of the stroke densities at the location of FS strokes is statistically different from the distribution of the maximum stroke densities. These conclusions demonstrate that fires are not starting at the location of the local maximum in lightning density.

A WMW test reveals a statistically significant difference between the distributions of rain rates (24-h rainfall total) at the location of the FS flash and values of maximum rain rate (24-h rainfall). The Z-scores are $-12.53$ ($-12.06$), and the p-values are $2.49 \times 10^{-36}$ ($6.49 \times 10^{-34}$). Both Distance B and the distance from the FS flash to the maximum 24-h total rainfall (Distance C) exhibit a relatively large standard deviation (Table 2). Thus, the results do not suggest the use of a typical distance for potential operational use. Compared to Distance A, fires are initiated further from the areas of maximum precipitation, in terms of both rain rate and 24-h accumulated rainfall (Table 2). Of the 95 fires, 79 (83.2%) contain FS flashes located > 4 km from the maximum rain rate and 4 (4.2%) have FS flashes located < 2 km away. Similarly, 88 (2) fires were located > 4 (< 2) km from the area of maximum rainfall.

*3.4. Case Studies*

Three case studies are evaluated in this section to highlight some of the key observations of the study. These fires include: The Parker 2, Lizard, and Bridge Creek Fires. The Parker 2 Fire demonstrates fire ignition in a grid cell containing a small stroke density rather than in the nearby grid cells exhibiting larger densities. The Lizard Fire illustrates that its FS flash is located where precipitation rates are small, while the Bridge Creek Fire shows that stroke densities typically remain small throughout the entire domain, as well as recognizing that land-surface conditions play an important role in quantifying LIW potential.

3.4.1. Parker 2 Fire

The Parker 2 Fire was ignited by five possible strokes between 1 and 2 AM on 25 July 2017 in Modoc County, CA, as a line of thunderstorms passed over the area (Figure 6). The fire burned about 31 km$^2$ [31]. It is an example of a long holdover event since the fire was not detected by satellite until 3 August, nine days after the storm passed. The storm system on 25 July was the only nearby storm within 10 days prior to fire ignition; therefore, it was the likely storm that caused the fire.

Locations of the FS strokes are superimposed on WSR-88D radar imagery in Figure 7a–c. The first stroke was a relatively strong –CG single stroke flash with a peak current of $-36$ kA (Figure 7a). Its associated 2-min rainfall rate was 4.2 mm h$^{-1}$, and the total 24-h rainfall was 17.6 mm at the FS location. Next, was a –CG multi-stroke FS flash with a multiplicity of five, but only three of its five strokes met the criteria (Section 2.2) to be denoted as fire starters (Figure 7b). Rainfall rates of these FS strokes were 11.375, 5.775, and 13.35 mm h$^{-1}$, and the 24-h total rainfall at each of these locations was 17.9 mm. The variability in the rain rates of the previous three strokes resulted from different ground stroke points, which placed these ground strike locations in differing grid boxes of the MRMS data.

Note the strong horizontal reflectivity gradient where these strokes touched the ground (Figure 7b). On the other hand, the 24-h rainfall totals were the same at all three ground strike points. The fifth and final potential fire starter was a weak –CG single stroke flash with an $I_p$ of -11.4 kA (Figure 7c). Its location was near an area of strong reflectivity, with a rainfall rate of ~1.4 mm h$^{-1}$ and a 24-h precipitation of 16.8 mm.

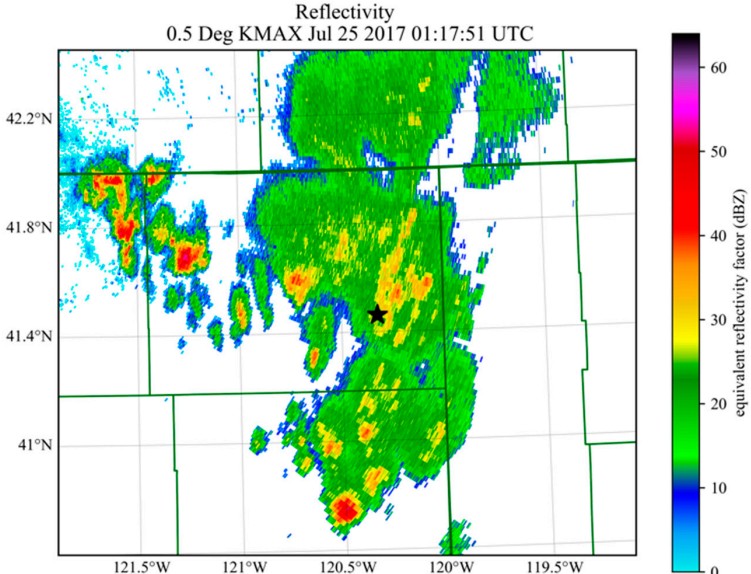

**Figure 6.** Base reflectivity (dBZ) from the WSR-88D radar in Medford, OR showing the storm system passing over northeast California that caused the Parker 2 Fire [45]. The black star denotes the satellite-derived fire start location nine days later. The time shown is near the time of one of the FS strokes' discharge.

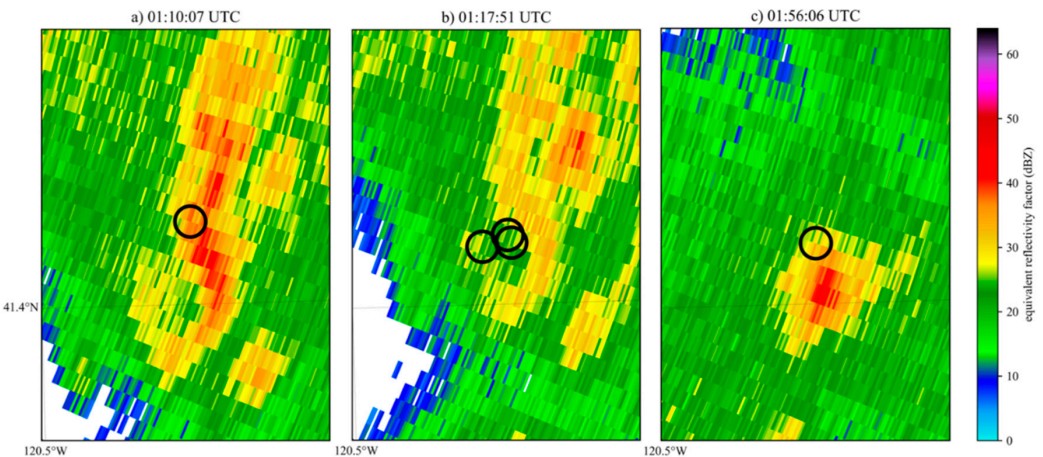

**Figure 7.** Base reflectivity (dBZ) from the WSR-88D radar in Medford, OR showing FS stroke locations (circles) near the times of their discharge at (**a**) 0110 UTC (**b**) 0117 UTC and (**c**) 0156 UTC 25 July 2017 [45].

The most notable feature of the Parker 2 Fire is the variability of stroke densities in its 20 km by 20 km domain. Although the greatest stroke densities are located near three of the domain's boundaries (Figure 8), the fire began in an area of smaller densities (2 strokes km$^{-2}$ or less). The maximum stroke density is 10 strokes km$^{-2}$, located ~6.7 km west of the fire. When flash density is analyzed, the maximum flash density within the domain is only 3 fl km$^{-2}$.

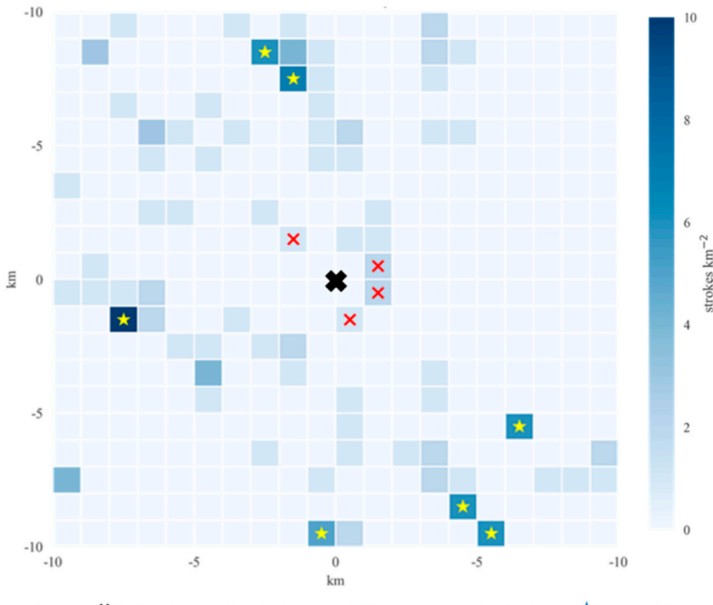

**Figure 8.** Stroke densities (strokes km$^{-2}$) within the 20 km by 20 km domain centered on the fire start location of the Parker 2 Fire from 0010 UTC 25 July 2017 through 0256 UTC 25 July 2017. The square boxes represent 1 km$^2$ grid cells with their stroke densities shaded accordingly.

### 3.4.2. Lizard Fire

The largest fire in the trio of case studies is the Lizard Fire which burned ~62 km$^2$ [31]. A thunderstorm on 7 June 2017 (Figure 9) near Arizona's Dragoon Mountain Range produced a lightning strike that caused the fire. The FS flash was negative and contained a single return stroke that was detected about 900 m from the satellite-derived fire start location (Table 3). It was not until the following day (8 June) that the fire was detected by satellite.

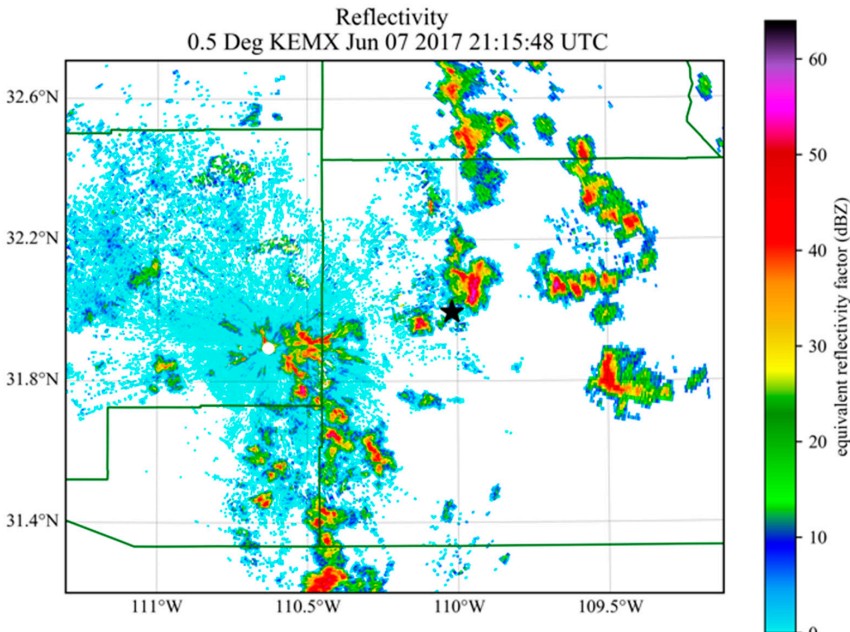

**Figure 9.** Base reflectivity (dBZ) of multiple cells from the WSR-88D radar in Tucson, AZ [45]. The black star denotes the location where the Lizard Fire started near the time of the lightning strike that caused the fire.

**Table 3.** CG flashes occurring the same day as the Lizard Fire initiation and their corresponding precipitation measurements, with the FS flash highlighted.

| Time (UTC) | Lat | Lon | Multiplicity | $I_p$ (kA) | Rain Rate (mm h$^{-1}$) | 24−h Rainfall (mm) |
|---|---|---|---|---|---|---|
| 20:39:40 | 32.018 | −109.943 | 2 | −12.5 | 12.3 | 27.225 |
| 20:40:41 | 32.017 | −109.946 | 3 | −24.3 | 7.7 | 31.6 |
| 20:41:35 | 32.019 | −109.945 | 4 | −37.2 | 5 | 31.6 |
| 20:42:14 | 32.018 | −109.959 | 4 | −20.9 | 0.95 | 18.225 |
| 20:42:49 | 32.040 | −109.950 | 1 | −2 | 0.075 | 34.65 |
| 20:43:26 | 32.024 | −109.944 | 2 | −16.3 | 130.475 | 27.225 |
| 20:50:45 | 32.050 | −109.964 | 4 | −42 | 51.05 | 22.3 |
| 21:07:15 | 31.989 | −109.989 | 1 | −26.5 | 0.1 | 0 |
| 21:09:29 | 31.989 | −109.992 | 8 | −18.9 | 0.1 | 0 |
| 21:12:30 | 32.021 | −109.947 | 3 | −22.4 | 98.8 | 28.4 |
| 21:15:05 | 31.995 | −110.008 | 1 | −20.7 | 0 | 0.1 |
| 21:16:25 | 32.044 | −109.945 | 5 | −18.3 | 147.4 | 42.7 |

The Lizard Fire was associated with 11 NFS flashes. Their corresponding precipitation quantities are listed in Table 3 while details of the FS flash are highlighted in Table 3. In terms of rainfall, the mean rate of the NFS flashes was 41.27 mm h$^{-1}$ while the 24-h total rainfall was 24 mm. The range of the NFS precipitation rates was 0.1–147.4 mm h$^{-1}$. One should note that the FS flash was located where there was little to no total rainfall. Radar reflectivity (Figure 10) reveals that the FS flash is located within low reflectivity, while the NFS flashes are generally located within higher reflectivity. This case illustrates, similar to 96% of the wildfires in this study, that CG flashes in an area of low precipitation rates have the potential for wildfire initiation.

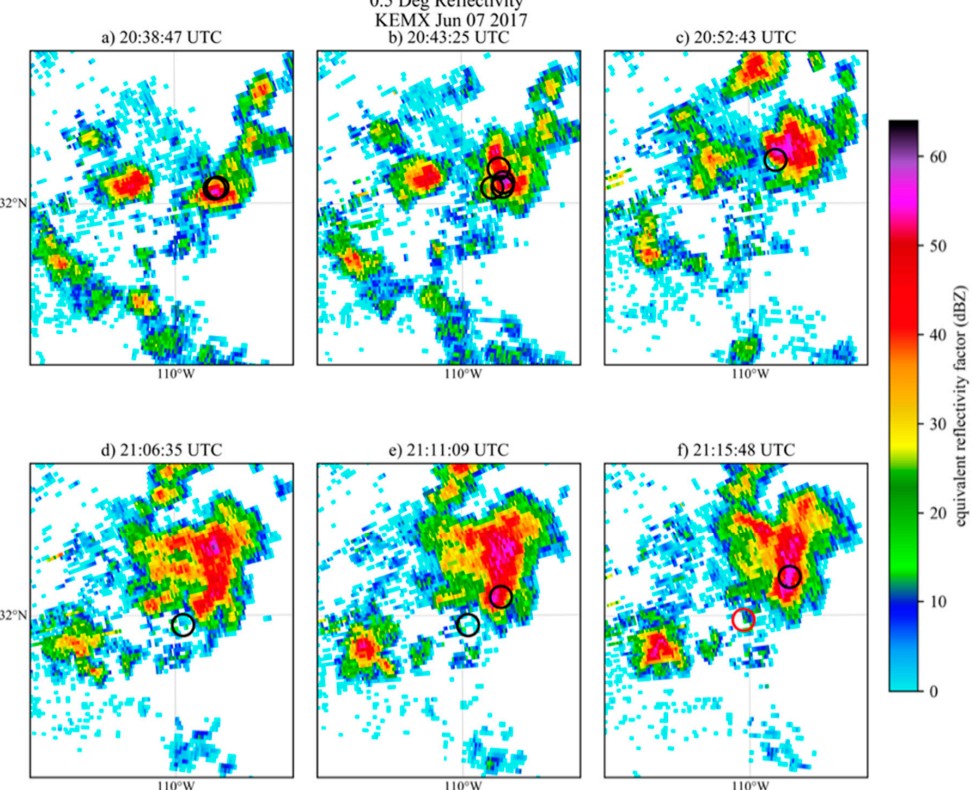

**Figure 10.** Base reflectivity (dBZ) at Tuscon, AZ superimposed with CG flashes that occurred during a 2-h period starting at 2038 UTC 7 June 2017 [45]. Black circles represent the NFS flashes, and the red circle denotes the single FS flash of the Lizard Fire.

### 3.4.3. Bridge Creek Fire

The Bridge Creek Fire was started by a lightning flash from a cell on 8 August 2017 (Figure 11). The fire first was detected by satellite on the following day (9 August 2017) near the Colville Reservation in northeast Washington. Approximately 15 km$^2$ were burned, which is considered relatively small in the wildfire community [31]. A negative, single stroke flash with $I_p$ of −14.4 kA was detected at 2332 UTC ~249 m from the fire's location. This was the only FS flash assigned to this fire. A local maximum rainfall rate was located east of the FS flash, and a local maximum in total rainfall was located east and northeast of the FS flash (Figure 12). The lightning-initiated strike is outside the area of maximum rain rate and total rainfall. The distance from the lightning-initiated flash to the maximum precipitation rate is 7.3 km, while the distance from the FS flash to the maximum flash density (3 strokes km$^{-2}$) is 5.50 km, just outside the area of maximum 24-h rainfall (Figure 12b). The fire initiated in an area of little rainfall, both at the time of the lightning strike and the encompassing 24-h period. In fact, both the rain rate (mm h$^{-1}$) and total rainfall (mm) were zero at the FS flash location. Therefore, the lightning-initiated strike of the Bridge Creek Fire was located outside of the area of maximum precipitation, and outside the grid cell with maximum stroke density. Two additional NFS flashes were observed outside areas of precipitation (−2.2 kA and −24.6 kA). Without land surface information, it is unclear why these flashes did not start additional fires.

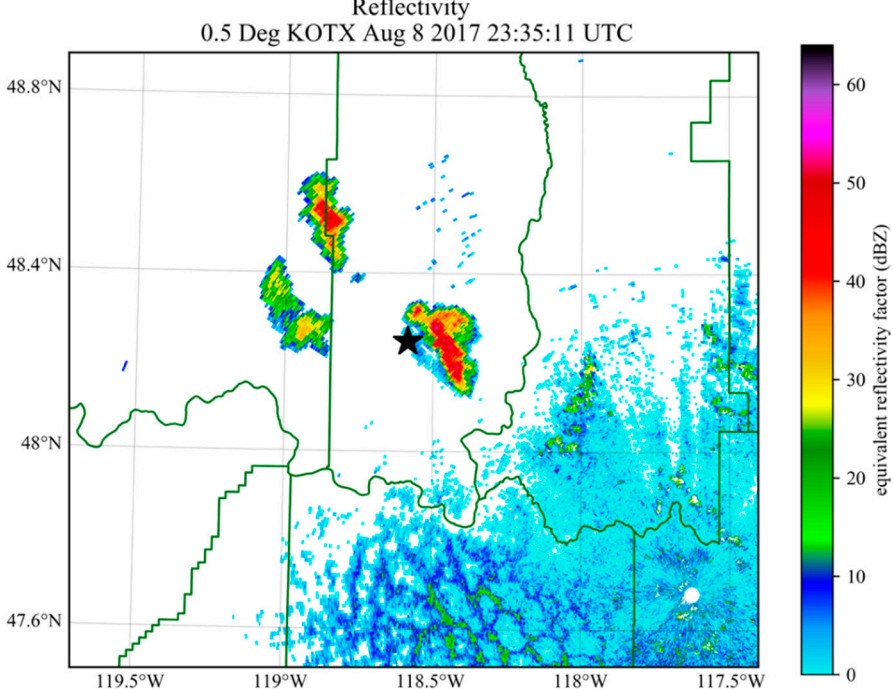

**Figure 11.** Base reflectivity (dBZ) from the WSR-88D radar in Spokane, Washington [45]. The black star denotes the location where the Bridge Creek Fire started. The time shown is near the time of the FS stroke's discharge.

The Bridge Creek Fire was associated with small flash densities that are representative of the typical flash densities of wildfires within the database of this study. The single stroke FS flash was located within a grid cell with a stroke density of 1 stroke km$^{-2}$. Unlike the Parker 2 case study, the stroke densities do not exceed 3 strokes km$^{-2}$ anywhere in the domain (Figure 13). This example reveals that not all dry lightning flashes ignite wildfires. Surface vegetation and soil moisture conditions play a vital role in quantifying LIW potential [1,46].

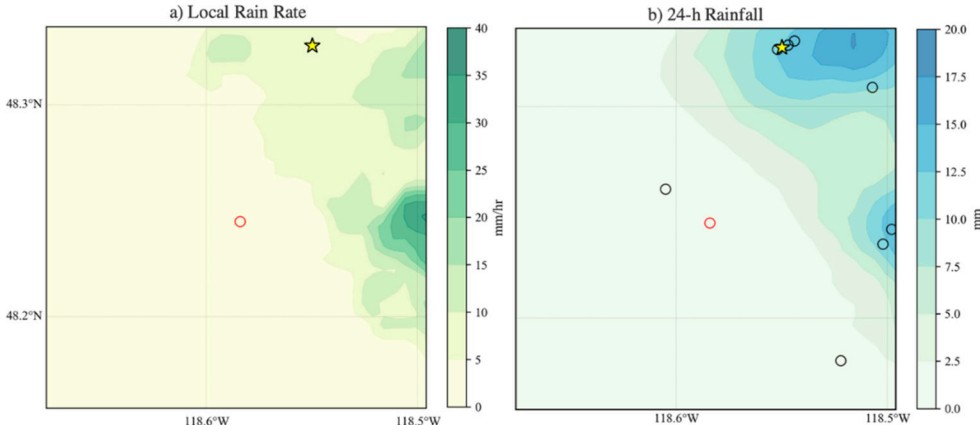

**Figure 12.** Plots of (**a**) rate rain (mm h$^{-1}$) at 2332 UTC 8 Aug 2017 and (**b**) 24-h rainfall (mm) from 1200 UTC 8 August 2017 to 1200 UTC 9 August 2017 over the 20 km by 20 km domain centered on the fire start location of the Bridge Creek Fire. The yellow star denotes the location of maximum stroke density, and black (red) circles represent the NFS flashes (FS flash).

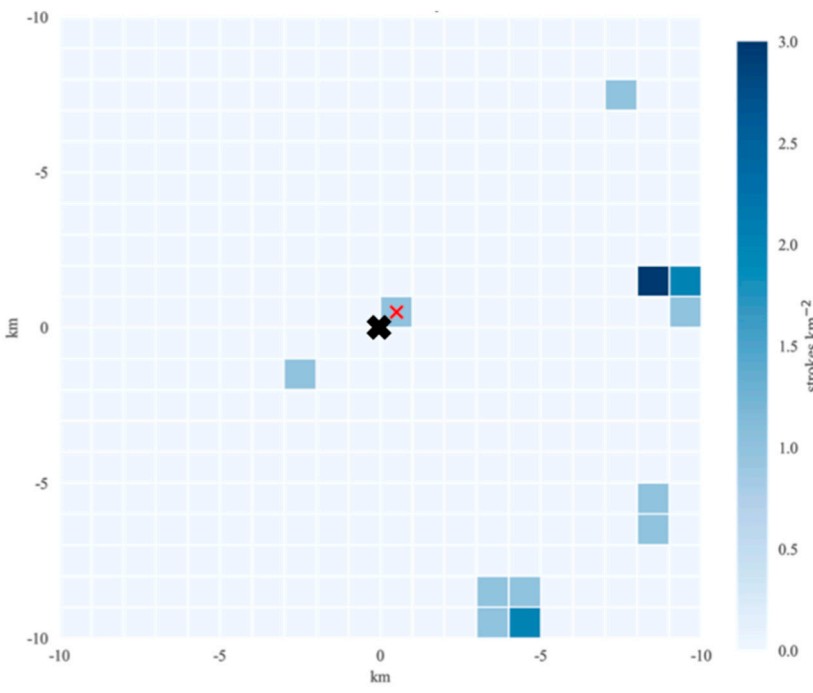

**Figure 13.** Stroke densities over the 20 km by 20 km domain centered on the fire start location of the Bridge Creek Fire from 2233 UTC 8 August 2017 through 0033 UTC 9 August 2017. Square boxes represent 1 km$^2$ grid cells, with corresponding stroke densities shaded accordingly.

### 3.5. Comparisons with Previous LIW Analyses

The present analysis continues to indicate that flash density at the locations of fire starts is smaller than the previously postulated minimum flash density of 5 fl km$^{-2}$ cited by Fuquay et al. [4] to ignite various fuels. Of the 95 events examined here, none exhibited a flash density exceeding 5 fl km$^{-2}$, and 83 of the 95 wildfires only had a single lightning flash within 2 km of the fire start location (Figure 3). These results continue to support the work of Schultz et al. and Sopko et al. [12,14] in that single flash km$^{-2}$ events are more likely to produce a LIW than multiple flash km$^{-2}$ events.

Daily precipitation totals in the present study also support the previously reported small 24-h precipitation totals at the locations of FS flashes [14,15]. The 24-h rainfall values observed here aligned favorably with the results of Sopko et al. [14] with the median rainfall of the entire FS dataset at 2.7 mm during a 24-h period at the FS flash locations. Part of the reason for the greater 24-h precipitation totals at FS flash locations in Sopko et al. [14] and the present study compared to Vant-Hall et al. [15] is likely due to the smaller grid spacing of the precipitation data (4 km in Sopko et al. [14], 1 km the present study, versus 0.1 deg (~10 km) in Vant-Hall et al. [15]).

The present analyses also support the observations of Sopko et al. [14] that holdover events are difficult to detect. The 2-min precipitation rate and 24-h precipitation totals for holdover events were more similar in magnitude to the NFS flash database using the Z-scores and *p*-values as metrics for similarity. This was most evident when comparing the 30 random samples of 24-h precipitation total from NFS flashes with the 2–10 day fire start population. In eight random sample comparisons of the 30 that were generated, Z-Scores were much less than 1.96. This denotes that the values from the FS dataset and random sample of the NFS flash population were similar and that both 2-min precipitation rate and 24-h precipitation total were not unique discriminators in some of the samples tested. The relatively low Z-Scores suggest that the populations were statistically similar. Thus, the easier tasks continue to be the events where lightning is observed and the fire is detected within one day. However, more research is needed to understand holdover events, and how the land surface and boundary layer allow smoldering for two or more days before full fire breakout occurs.

## 4. Conclusions

This study examined the characteristics of lightning strikes that caused 95 wildfires in the western U.S. during 2017, using VIIRS satellite-derived products to locate natural-wildfires and NLDN data to determine the lightning flashes that contributed to the fire starts. The average flash density at the location of fire starts was 1.1 fl km$^{-2}$. The majority (89%) of the flashes exhibited negative polarity, and 66% of these were single stroke flashes.

FS flashes corresponded to smaller 2-min precipitation rates and smaller 24-h precipitation totals than those observed at the locations of NFS flashes. It was also demonstrated that fires that were detected within 1 day of lightning occurrence had smaller precipitation rates at the time of the flash and smaller 24-h total precipitation than fires detected 2–10 days after lightning occurrence.

Meanwhile, the median separation between FS flash locations, the maximum stroke density, maximum precipitation rate and 24-h precipitation total were 5.3 km, 7.4 km, and 10.2 km, respectively.

The new approach of the characterization of the NFS flash population and separation of holdover events from events that immediately produce a satellite detectable fire, as presented in this study, shows an expressive improvement in the determination of lightning-initiated wildfires events, providing a particular innovation in the methodology.

Nevertheless, future research should utilize a larger database to increase wildfire diversity. Additional geographic regions should be studied to consider regional differences in flash density and precipitation rate. The utilization of NOAA's Geostationary Lightning Mapper (GLM) data [47,48] in addition to NLDN data will be valuable since GLM provides a metric of flash energy related to continuing current [7]. Additionally, land-surface information must be coupled with the atmospheric data studied here to include the effects of soil moisture, land-surface type, and fuel moisture on fire ignitions (e.g., Sopko et al. [14]). This coupling of information will provide a more thorough method for predicting lightning-caused wildfires.

**Author Contributions:** Conceptualization, B.R.M. and C.J.S.; Data curation, B.R.M.; Formal analysis, B.R.M. and C.J.S.; Funding acquisition, C.J.S. and H.E.F.; Investigation, B.R.M.; Methodology, B.R.M.; Project administration, C.J.S.; Resources, B.R.M. and C.J.S.; Supervision, C.J.S. and H.E.F.; Validation B.R.M.; Visualization, B.R.M.; Writing—original draft, B.R.M.; Writing—review and editing, B.R.M., C.J.S. and H.E.F. All authors have read and agreed to the published version of the manuscript.

**Funding:** This research was funded by the Advanced Computing for Earth Sciences (ACES) program and by Florida State University.

**Acknowledgments:** The authors would like to thank Jon Case and Tim Lang at NASA SPoRT for their help coding scripts in Python. Also Mark Bourassa and Guosheng Liu at Florida State University for a thorough review during the research process. Lastly, the authors would like to thank Neil Dixon, Michael Stroz for the improvement of the manuscript. Lightning data are available through Vaisala Inc.

**Conflicts of Interest:** The authors declare no conflict of interest.

## Appendix A

**Table A1.** Statistics of 2-min rain rate at the closest MRMS time for each FS and NFS flash.

| | FS Flashes | NFS Flashes |
|---|---|---|
| | 2-min rain rate (mm h$^{-1}$) | |
| 25th percentile | 0.43 | 1.65 |
| Median | 1.69 | 4.65 |
| 75th percentile | 4.66 | 15.55 |
| Mean | 4.33 | 12.36 |
| Standard deviation ($\sigma$) | 7.41 | 17.73 |
| Z-Score | −5.58 | |
| WMW test *p*-value | $1.21 \times 10^{-8}$ | |

**Table A2.** Statistics of 24-h rainfall totals at FS and NFS flash locations.

| | FS Flashes | NFS Flashes |
|---|---|---|
| | 24-h QPE storm total (mm) | |
| 25th percentile | 1.68 | 4.13 |
| Median | 2.94 | 8.55 |
| 75th percentile | 6.89 | 15.46 |
| Mean | 5.34 | 10.63 |
| Standard deviation ($\sigma$) | 6.09 | 8.30 |
| Z-Score | −7.18 | |
| WMW test *p*-value | $3.58 \times 10^{-13}$ | |

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
