# Peer review of "Flash Characteristics and Precipitation Metrics of Western U.S. Lightning-Initiated Wildfires from 2017"

_fire, doi:10.3390/fire3010005_

Round 1

Reviewer 1 Report

What is the spatial resolution of VIIRS sensor and what is the potential effect of this resolution on the presented results?

In the paper line 129 it says: If multiple flashes were located within 2 km of the fire start location, all of them were retained as FS flashes. But doesn’t this mean that your distribution of FS flashes contains a number of actually NFS type. Authors assigned 142 FS flashes to 95 fires, which means that about 1/3 of the 142 flashes did not really start a fire. Can authors do an estimate of the range of uncertainty of their results due to this extra flashes/strokes? E.g. Authors could do the analysis for the subset of fires with a single FS flash in the 2km range and check if this gives similar results as the total of the 95 fires.

Fig.2: The green color of the FS line is hard to see (at least on my printer) - I suggest to use either another color or another symbol to make it more distinguishable.

Line 252: Authors say “Approximately 42% (60/142) of wildfires in the present study…. – shouldn’t this say 65% (60/92) of wildfires …., because 142 is the number of FS flashes in the 95 fires? – Please check.

Author Response

1) I mentioned in the manuscript that the spatial resolution of VIIRS is 375-m. I did add a remark about the potential limitations of this resolution. I placed it at the end of section 2.1.1.

2) This was something I thought about doing, however, one could argue that there is a possibility that more than one flash/stroke could have started multiple small fires (but very close spatially) that merged into the larger fire that was then sizable to be detected via satellite. In other words, if a single wildfire was assigned 3 FS flashes, then technically all 3 of those could in fact be "true" fire starter flashes. Contamination of "actual" non-fire starter flashes is inevitable to some degree, however; how much contamination is unknown. For all we know, there could be zero contamination (probably unlikely), but because it is possible that all the potential FS flashes could have started the fire (in the cases where a fire had more than 1 FS flash) then I think it would be hard to justify doing an analysis on a subset where each fire would be assigned only 1 FS flash.

3) Fixed! The green dot should appear much more brighter and visible now.

4) Fixed - this should have said "42% (60/142) of FS flashes", not wildfires.

Reviewer 2 Report

The paper is an interesting study of Lighting initiated wildfires. The approach is simple but very effective.

See the comments in the pdf attached. 

Reviewer 3 Report

This study seeks to relate lightning stroke characteristics to lightning initiated wildfires. The authors conclude that dry conditions typically prevail when wildfires are initiated by lightning.

The introduction and methodology for this study is adequate. The analysis presented in this study is sound. The presentation needs work and could be published after major grammatical and structural revisions are done. I have enclosed all comments and suggestions in the PDF.

Author Response

Referring to the original PDF that was sent:

Lines [38-42]: Shortened/combined details to make more concise.

Line [48]: The first sentence now ends with a semi-colon to lead into the following sentence in order to correct the sentence fragment issue. Hopefully this makes it more clear now.

Line [293]: Sentence was removed since the following sentence does get the point across.

Line [309]: Sentence removed.

Line [334]: First half of that sentence was removed; those gory details were unneeded.

Line [383]: Sentence removed. Again, details are too specific and unneeded in this case.

**All other smaller revisions (e.g., removal of single words, grammar issues, number notations, rewording statements within sentences, etc.) that I did not explicitly refer to here were revised exactly according to the suggestion made available by the reviewer.

Round 2

Reviewer 2 Report

#----DETAILED COMMENTS----#

#According to ACS Style Guide that MDPI uses as model, when the author’s name may be part of the sentence, you should put the reference number after the names, as showed below. However, remembered that you must use a square bracket “[ ]” instead of parenthesis “()”.

“Jensen (3) reported oscillation in the reaction of benzaldehyde with oxygen.”

In this way, I suggest you to correct these in your text.

Issues found in paragraphs [56]; [59]; [195]; [282]; [440; [443]; [447], [449], [450], [451; [484]

#[75] Correct the time series to “April to October” to match with your Results. Because now you show the reasons to exclude the April.

#[177] Is the maximum precipitation rate at 2-min? Include the “2-min”, just to be more explicit

#[206] Put the Figure 3 after you call it in the text.

#[226 and 231] I guess you can put again the reference “[17]” after you cited the thresholds.

#[297] “[…] rate in section 3.2.1” you are still in this section. You can remove that part, and make something as “Similar to the 2-min precipitation rate, we found a statistically significant differences ….”

[354]The reference 46 is a library to work with Radar in Python. Is these reference necessary in the legend ?

#Conclusions

I reorder your paragraphs in the conclusion, to make your ideas flows properly. Check if you agree with it.

“This study examined the characteristics of lightning strikes that caused 95 wildfires in the western U.S. during 2017, using VIIRS satellite-derived products to locate fires, and NLDN data to determine the lightning flashes that contributed to the fire starts. The average flash density at the location of fire starts was 1.1 fl.km-2. Which the majority (89%) of the flashes exhibited negative polarity, and 66% of these were single stroke flashes.

FS flashes corresponded to smaller 2-min precipitation rates and smaller 24-h precipitation totals than those observed at the locations of NFS flashes. It also was demonstrated that fires that were detected within 1 day of lightning occurrence had smaller precipitation rates at the time of the flash and smaller 24-h total precipitation than fires detected 2-10 days after lightning occurrence.

Meanwhile the median separation between FS flash locations, the maximum stroke density, maximum precipitation rate and 24-h precipitation total were 5.3 km, 7.4 km, and 10.2 km, respectively.

The new approach of characterization the NFS flash population and separation of holdover events from events that immediately produce a satellite detectable fire, presented in this study, shows an expressive improvement in the determination of lightning initiated wildfires events, providing a particular innovation in the methodology.

Nevertheless future research should utilize a larger database to increase wildfire diversity, and additional geographic regions also should be studied to consider regional differences in flash density and precipitation rate. As the utilization of NOAA’s Geostationary Lightning Mapper (GLM) data in addition to NLDN data will be valuable since GLM provides a metric of flash energy related to continuing current. As well a land-surface information must be coupled with the atmospheric data studied here to include the “

Reviewer 3 Report

Please view the enclosed PDF of reviewer comments.

Author Response

line [122] both sentences were modified exactly as suggested

line [151] was indented

line [159] fixed grammar

line [167] fixed grammar

line [191-192] "2 km" was modified for consistency

line [200] sentence was removed

line [322] the word "now" was removed

line [421] figure 11's time/date format was changed for consistency (as well as other figs with the same issue) - all figs should now have the same time/date format

line [443] sentence structure/formatting changed as suggested